# Telemedicine and Its Past, Present, and Future Roles in Providing Palliative Care to Advanced Cancer Patients

**DOI:** 10.3390/cancers14081884

**Published:** 2022-04-08

**Authors:** Michael Tang, Akhila Reddy

**Affiliations:** Department of Palliative, Rehabilitation, and Integrative Medicine, MD Anderson Cancer Center, The University of Texas, Houston, TX 77030, USA; mjtang@mdanderson.org

**Keywords:** telemedicine, palliative care, cancer, pain management, opioids, health care delivery

## Abstract

**Simple Summary:**

The health care delivery model has dramatically changed due to the emergence of the global pandemic coronavirus disease (COVID-19). This can be seen in the innovative adoption of telemedicine in the delivery of palliative care to patients with advanced cancer. We provide an update on the adoption, delivery, benefits, and challenges faced in this model of health care delivery.

**Abstract:**

The landscape of healthcare delivery has considerably changed due to the emergence of coronavirus disease 2019 (COVID-19). This is nowhere more evident than in the care of advanced cancer patients receiving palliative care. This population is susceptible to the severe complications of COVID-19, and immediate measures had to be taken to ensure their safety. Thus, the adoption of telemedicine as a health care delivery model emerged. This model provides many benefits, such as improved access to care while maintaining social distancing; however, there exist challenges to this model, including health care disparities, reimbursement, and monitoring of opioids in high-risk populations. This narrative review provides an overview of the unique benefits and barriers of telemedicine in palliative care patients.

## 1. Introduction

Telemedicine, or the use of telecommunications to provide health services, is a technology that has been long studied to help increase access to health care [1,2]. Its origins can be traced back to the United States (U.S.) Civil War, in which the telegraph was used to help transmit the medical supply needs of soldiers in the field [3]. Early mentions of the potential of telemedicine providing remote care are seen in a case report in the Lancet 1879, in which a physician provided care to an infant with the use of a phone [4]. As the technology grew, the vast potential of telemedicine took form. Willem Einthoven (1860–1927), a Dutch physician, transmitted heart sounds with the use of a galvanometer and the emerging telephone in 1905 [5]. In 1920, one of the first documented uses of telemedicine by a service occurred at Haukeland Hospital in Norway, in which radio links were used to provide health care support to ships at sea [6]. From its infancy to today, the potential and use of telemedicine has grown dramatically.

The current landscape of health care delivery changed due to the emergence of coronavirus disease 2019 (COVID-19). Healthcare providers encountered unprecedented challenges in providing care to patients. This included the need to accommodate social distancing, caring for those in the hospital who are most vulnerable to communicable diseases and providing support to families who cannot be with their sick loved ones [7,8,9]. In addition, health care systems worldwide have been under tremendous strain due to the increasing number of patients with COVID-19 [10]. Telemedicine has emerged as a platform to provide care for patients while helping to ensure social distancing not only among patients, but also between patients and medical teams [10]. During the first quarter of 2020 in the U.S., the number of telehealth visits increased by 50% compared to the same months the prior year [11]. When the Center for Medicare and Medicaid Services telehealth waivers went into effect early in March of 2020, there was a 154% increase in telemedicine visits as compared to the same period in 2019 [11].

Palliative and supportive care programs worldwide have played a vital role in the COVID-19 pandemic response. Many programs have utilized telemedicine to help continue providing support to patients, families, caregivers, and treating teams [8]. This health care delivery platform allows continuity in providing effective symptom management, addressing goals of care, and conducting family meetings in a time where not all involved can be physically present together [8]. Despite the great promise that telemedicine has in palliative medicine, there are challenges ahead. For instance, there are barriers to telemedicine implementation, including resource-limited settings, and healthcare and health literacy disparities among different populations [10,12,13]. This narrative review article sets out to describe the past and present use of telemedicine in the field of palliative care, along with discussing challenges that lay ahead for its continued use. Specifically, we shall discuss telemedicine use before the COVID-19 pandemic, the enormous increase in use during the pandemic with emphasis on palliative care, various challenges related to technology, regulations, reimbursement, conveying empathy, and managing non-medical opioid use, and conclude with focusing on the benefits of telemedicine in advanced cancer patients receiving palliative care.

## 2. Telemedicine and the Management of Chronic Diseases

To better understand telemedicine’s role in providing palliative care to patients with advanced cancer, its role in the management of chronic diseases must be discussed. Much of what emerged in palliative medicine and its use of telemedicine is a direct result of the studies conducted in the past. This section of our review article is not meant to be an all-encompassing examination of the early landmark articles in telemedicine but rather to explore how early trials were designed and the outcomes they demonstrated.

One of the early landmark studies in the management of chronic diseases with telemedicine took place in California in the late 1990s at Kaiser Permanente [14]. The paper’s authors evaluated a videoconferencing system that allowed patients and nurses to speak in real-time [14]. The patients had several chronic diseases, including congestive heart failure, chronic obstructive pulmonary disease (COPD), diabetes, and cancer [14]. The intervention group received video conferences to talk to a nurse in real-time along with home visits [14]. The control group received home visits along with phone calls [14]. Some of the results published showed that while direct costs were higher in the intervention group, the total cost of care was lower in the intervention group when factoring in hospital, laboratory, pharmacy, physician visit, and emergency department visit costs [14]. The direct costs were higher due to the equipment that had to be purchased for video conferencing [14]. When looking at quality indicators, the group found no difference in patient perception on the quality of care provided between the two groups and confidence in their providers’ ability to assess their health conditions [14]. They found that those in the intervention group felt that the video visits were very convenient and allowed for timely access to the provider [14].

A couple of years later, in 2002, another group in California published a study evaluating nurse case-management telephone communications in patients with congestive heart failure who had just been discharged from the hospital [15]. The study was designed in that an intervention group received a phone call five days after a hospitalization for heart failure and thereafter, based on software recommendations [15]. Patients in the intervention group received an average of 17 phone calls at decreasing frequency over the six-month follow-up period (median 14 phone calls) [15]. Physicians were also updated on the patient’s progress through the software algorithm [15]. The study found that heart failure hospitalization rates were 45.7% lower in the intervention group than usual care at three months (*p* = 0.03), and 47.8% lower (*p* = 0.03) in the intervention group compared to usual care at six months [15].

To conclude, in 2012, a group in New York published a study looking at telemedicine interventions in a homebound older adult population with heart failure or COPD [16]. The group used a multi-faced model to manage the care of this population called the tele-HEART intervention, which included the following: in house assessments and setup, education on the monitoring devices, ongoing care by a nurse specialist, evaluation and management of heart failure, and COPD and comorbid depression, use of an electronic health record and tracing tools [16]. The intervention group would enter data into the system, which nurses reviewed daily [16]. Over a three-month intervention period, the intervention group was contacted an average of 18 times [16]. The authors found that the intervention group had significantly improved depression scores using the PHQ-9 and the CES-D [16]. The intervention group also had significantly fewer visits to the emergency room and an observed trend toward fewer hospitalizations [16].

## 3. Telemedicine and Its Role in Palliative Medicine

Before the COVID-19 pandemic, relatively few studies looked at the potential impact of telemedicine in the care of patients who receive palliative care. The role of this section is to review some studies conducted before the beginning of the pandemic. One of the first studies we examined was published in 2013 [17]. The authors evaluated telemedicine to improve access to a specialist multidisciplinary palliative care team for rural cancer patients [17]. The study took place at a tertiary cancer center in Edmonton, Canada. For patients who reside in rural areas, systemic chemotherapy may be received through a network of associate and community centers under the care of internists and family physicians [17]. In 2007, the Ministry of Health of Canada approved a pilot program grant to examine the use of video conferencing for palliative care and palliative radiation therapy for patients in northern Alberta, Canada [17]. The study began in 2008 and was completed in 2011 [17]. Visits occurred via telemedicine with the palliative care interdisciplinary team [17]. After the visit, the patient stepped out of the room, and a care conference was held with the rural care team at the center [17]. The patient returned to discuss recommendations. Eighty-eight patients were referred to the virtual clinic, and 44 were eligible to participate [17]. The researchers estimated that distance savings per visit for the patients was 471.13 km [17]. The average time and cost savings per visit were estimated to be at 7.96 h and CAD 192, respectively [17]. Approximately 16% of patients estimated their cost would have been over CAD 500 to visit the main hospital in Edmonton, Canada [17]. Four patients reported that they would not have been able to afford to travel to the Cancer Center in Edmonton [17].

Another study that showed the potential of telemedicine in the management of symptoms of advanced cancer patients was published in February of 2016 [18]. This study evaluated a systematic web-based collection of patient-reported symptoms during chemotherapy treatments with automated alerts to clinicians for severe or worsening symptoms [18]. This occurred as a single-center trial in which 766 patients with solid tumors diagnosed with advanced cancer were enrolled [18]. The intervention group, which was the group that received the systematic web-based reporting system, was compared to usual care, which was regularly scheduled for face-to-face visits [18]. This study found that the web-based intervention group had fewer patients visiting emergency rooms, improved health-related quality outcomes, decreased hospitalizations, and improved survival [18].

A study published in 2015 and conducted in Rio de Janeiro, Brazil, aimed to evaluate telemedicine to monitor symptoms in patients with advanced cancer [19]. It was a single-center study conducted from 2011 to 2013 [19]. The authors found that those monitored via telemedicine had lower symptoms scores than those seen only in person [19]. The study also noted the benefits of seeing the patients in their own households. The authors mentioned that clinical conditions such as bedsores, edema, and dyspnea were noted in patients, along with aspects related to the patient’s comfort [19]. For instance, the clinicians could observe movement in the house and the patient’s ability to walk to the computer [19]. The interviewers were also able to ask questions about the patient’s doubts about the treatment, prognosis, social rights, and advanced care planning [19].

Not all studies have shown a positive effect with telemedicine in a palliative care population. Hoek et al. examined patient-reported symptoms by comparing a telemedicine intervention of weekly visits versus usual care, defined as routine visits determined by the complexity of the patient’s symptoms and stage of their underlying condition [20]. Those who received weekly visits reported a higher symptom burden compared to usual care, based on the Edmonton Symptom Assessment Scale [20]. The authors felt that weekly visits and frequent assessments might inadvertently have caused increased attention to symptoms and a recall bias among the intervention group [20]. These findings must be explored further in future studies.

## 4. COVID-19 and Its Effect on Health Care Delivery in Palliative Medicine

The need for social distancing and patient safety during the COVID-19 pandemic has played a significant role in the emergence of telemedicine as a viable health care delivery model. As previously mentioned, the U.S. saw an increase of 154% in telehealth visits at the end of March 2020 compared to the same period in 2019 [11]. There were likely multiple factors that contributed to this, including the COVID-19-related policy changes and regulatory waivers that went into effect from the Centers for Medicare and Medicaid Services [11]. There was also a growing overall sentiment from the public to avoid seeking care because of concerns of exposure to COVID-19. An estimated 41–42% of U.S. adults surveyed reported having delayed or avoided seeking medical care during the pandemic because of fears of the infectious spread of the virus [21,22]. As a result, health care systems needed to find a way to transition to virtual care during the COVID-19 pandemic.

It was crucial for outpatient palliative and supportive care clinics to transition to telemedicine as soon as possible. The majority of the patients in supportive and palliative care clinics have advanced cancer or other chronic conditions and may be immunocompromised. It was critical to safeguard these patients from exposure to COVID-19, while continuing to help with their symptom management, access to opioids, and other supportive care needs. It was also essential to maintain access to counseling and other interdisciplinary services that a supportive care center can offer during a high-stress time of the pandemic. Many palliative medicine clinics rapidly transitioned to a virtual model of care to help continue to provide services to patients during the early months of the pandemic. Many such clinics, including ours, continue to use telemedicine in 2022 predominantly.

Our transition from in-person to telemedicine in March 2020 in response to the COVID-19 pandemic is an example of a successful process [23]. We conducted a retrospective chart review of 1744 consecutive patients seen between 14 February 2020, and 16 April 2020 [23]. The periods of interest were the four weeks before the transition to telemedicine, the one-week transition, and four weeks after the transition. Before the transition, 100% of the visits were in-person [23]. This decreased to 77% in-person visits during the transition week, and 13% in-person in the four weeks after the transition [23]. We found that in the four weeks after the transition, there was a significant decrease in walk-in/unscheduled visits and a considerable reduction in no-shows or missed appointments per day, [23] implying that telemedicine improved access to palliative care among our patient population. Moreover, we saw more patients in the four weeks after the transition to virtual care than before the transition [23].

Despite the transition to virtual care, our quality of care outlined by the completion of various supportive care assessments of our patients and the involvement of interdisciplinary team such as counselors did not differ significantly from before the transition. We maintained a consistent workflow in virtual care by continuing a similar pattern as our in-person visits. Many crucial assessments such as the Edmonton Symptom Assessment Scale (ESAS), the Eastern Cooperative Oncology Group Performance Status Scale (ECOG), the Cut-Down, Annoyed, Guilty, Eye-Opener (CAGE) questionnaire, and the Memorial Delirium Assessment Scale (MDAS) were successfully completed on video. Members of our interdisciplinary team, such as our counselors, pharmacists, and social workers, were able to join these telemedicine visits as needed to continue offering interdisciplinary palliative care to our patients [23]. Our clinic workflow for delivery of palliative care via telemedicine is shown in Figure 1.

Although the COVID-19 pandemic resulted in widespread adoption of telemedicine in palliative care, outreach psychological counseling to palliative care patients via telemedicine was previously shown to improve access to care and continuity [24]. Our prior experience with this outreach program helped our team accomplish a rapid and fruitful transition to telemedicine.

This rapid transition was also seen in other palliative care clinics worldwide. Another example was published in 2021 by authors at the Dana Farber Cancer Institute palliative care clinic [25]. In their paper, the authors reported a successful transition to a virtual health care delivery model [25]. It was easier for their team to reach out to patients to have proactive serious illness conversations, which took on increased importance in the setting of the COVID-19 pandemic [25]. Their pharmacy team was also able to contact patients and help manage complex medication regimens [25]. The clinic maintained the same number of patient contact hours before and after the transition [25]. Both of these settings demonstrate the ability of a palliative care outpatient clinic to transition from entirely in-person to virtual health care delivery models.

## 5. Barriers in Telemedicine

### 5.1. Webside Manner

There are challenges in implementing telemedicine as a health care delivery platform in the palliative care setting. One of the primary challenges is communication. Communication with patients and families, conveying empathy, clarifying the goals of care, and exploring their values and preferences for care is an integral part of delivering high-quality palliative and supportive care. Telemedicine poses unique challenges in communicating effectively, especially for palliative care teams. Identifying this unique challenge, Chua et al. published a paper detailing simple but highly effective modifications in communication that can bring empathy and compassion into telemedicine visits with our patients and their families [26]. The purpose was to introduce a suitable “webside manner” during telemedicine visits, as shown in Table 1 [26]. Suggestions to improve webside manner include the proper position of the camera and clinician, eye contact, normalizing the initial awkwardness, minimizing overtalking, avoiding prolonged silences, body positioning to lean in slightly, using gestures appropriately, navigating technical difficulties, appropriately using an interpreter where needed, involving interdisciplinary teams, and summarizing and outlining the next steps [26]. It is now essential to incorporate the training for effective telemedicine as early as in medical school and refine it further in residency and fellowship programs.

### 5.2. Technology Barriers

Besides the challenges faced with effective communication during telemedicine, several technology-related barriers exist, especially in countries with low socioeconomic status [27]. This is particularly prevalent in developing nations with limited infrastructure and the inability to afford the technology among patients and clinicians [27]. As of 2020, nearly 3.6 billion people were without internet subscription access, while 93% of the world’s population lives within the radius of a cellular signal [27,28]. It is estimated that 2.9 billion people live offline in the developing world [29]. Approximately 43% of households do not have Internet access at home around the world. When looking at Africa and Asia, approximately 71.8% and 51.6% do not have access to the Internet [27]. Even within developed nations, some disparities exist concerning access and availability of the Internet. In 2019, the U.S. Federal Communications Commission (FCC) estimated 19 million Americans do not have broadband service [30]. Approximately 25% of those who live in rural communities do not have access to these services [30]. It should also be noted that although one has access to the service, the service may not provide internet speeds that are required for effective audio-video communication [30].

Additionally, many elderly patients and families may not possess smartphones or computers. They may not have adequate knowledge of using these devices to participate in telemedicine video visits successfully [31]. In 2016, only two-thirds of U.S. adults over the age of 65 reported internet use. When considering broadband use, only half of those over the age of 65 reported having access to this technology [32]. With regards to knowledge gaps with the technology, there are estimates that 32% of older adults would not be able to participate in telemedicine [32]. In the palliative care setting, cancer patients who were male, Spanish-speaking, uninsured, and those who did not have an activated patient portal were less likely to utilize telemedicine [33].

### 5.3. Laws and Regulations

Another barrier to implementing this technology can be laws and regulations surrounding telemedicine and opioid prescribing. The relaxed rules for telemedicine during the COVID-19 pandemic allowed for the rapid transition to telemedicine. Advanced cancer patients frequently experience pain, and the majority of the patients seen in outpatient palliative care clinics, such as ours, receive opioids for cancer-related pain. Opioid prescribing has been scrutinized due to the opioid epidemic occurring within this pandemic [34]. Before the COVID-19 pandemic in the U.S., there were many laws in place to limit opioid prescribing via telemedicine.

An example of this can be seen in North Dakota, which prohibited prescribing opioids through telemedicine without exceptions [34]. Hawaii allowed opioid prescriptions via telemedicine only if the first visit was in-person [34]. The relaxed regulations around prescribing opioids via telemedicine helped many palliative care teams adopt telemedicine in their practice. However, it remains to be seen if these relaxed policies stay in place even after the pandemic. A reversal of any of these policies would be a setback for all patients, especially advanced cancer patients receiving palliative care and pain management.

Medical licensures and accompanying regulations are also a barrier to the widespread use of telemedicine in the U.S. [32]. Due to the COVID-19 pandemic, several states temporarily halted the need for out-of-state physicians to have an in-state medical license [32]. This allowed for the wide use of telemedicine beyond state lines [32]. However, many state medical licensing laws are reverting back to the pre-COVID-19 era [32].

### 5.4. Reimbursement

Along with relaxed regulations around telemedicine and opioid prescriptions, there were considerable changes in the reimbursement for telemedicine [35]. Telemedicine visits were historically reimbursed lower than in-person visits, making it a barrier for clinicians to offer virtual care to patients. Similar to the regulations around opioid prescribing, the reimbursement for virtual visits must remain favorable for patients and clinicians after the pandemic [35,36]. In countries such as the U.S., the future of telemedicine is highly dependent on policy decisions regarding reimbursement after the COVID-19 pandemic [37].

## 6. Nonmedical Opioid Use and Telemedicine

Cancer patients frequently exhibit behaviors of nonmedical opioid use and misuse of opioids [38,39]. Comprehensive interdisciplinary opioid stewardship programs like the Compassionate High Alerts Team (CHAT) established in supportive care clinics must be modified to be delivered virtually [40,41]. Additionally, patients exhibiting behaviors of nonmedical opioid use must be seen in-person, with virtual care allowed only after certain parameters have been met to demonstrate safety [40]. Within the COVID-19 pandemic, there have been studies that have shown increased misuse of opioids. For example, a paper published in Ontario, Canada, in 2021, looked at routine urine drug screens from 67 opioid agonist treatment clinics in Canada [42]. The authors conducted a chart review of patients who had visits between January 2020 and September 2020 [42]. They found the percentage of abnormal fentanyl positive tests and patients increased by 108% from April 2020 to September 2020 [42]. When analyzing the data, the authors found a disproportionate increase in the northern and southwestern portions of Ontario where there is a higher percentage of rural communities that face a barrier to access of care [42]. Our group has also published a report on monitoring high-risk patients during the COVID-19 pandemic through a risk mitigation strategy [40]. This involves an interdisciplinary team model and monitoring of high-risk behaviors [40]. In this model, those who would demonstrate high-risk nonmedical opioid use behavior would get telemedicine video visits every two to four weeks and an in-person visit every fourth week [40]. Urine drug screens would be performed on a random basis [40]. Those who have abnormal results on the urine drug screen would get in-person visits every one to two weeks, along with random urine drug screen monitoring with an opportunity to transition to telemedicine visits if compliance and adherence to the prescribed opioid treatment plan are established [40].

## 7. Benefits of Telemedicine in the Care of Patients with Advanced Cancer

There are numerous reported benefits of telemedicine in the care of patients with advanced cancer [32]. These include overall improved access to interdisciplinary oncological and palliative care and increased patient satisfaction [43]. Patients can have access to the oncologist and other subspecialties, along with members of the multidisciplinary team such as dieticians, social workers, and counselors. The patients can be in one location and access multiple services, reducing significant travel time and cost. This improved access to care in rural populations is especially remarkable. They often have poor access to timely healthcare, are diagnosed with cancer in later stages, with limited access to palliative care teams, and have higher cancer-related mortality. [32]. Telemedicine can also improve treatment adherence, improve continuity, enhance communication, and allow rural patients to participate in clinical trials. Telemedicine can significantly reduce missed appointments among advanced cancer patients [44]. Minimizing “no-shows” or missed palliative care appointments in advanced cancer patients is essential, as many patients visit emergency rooms in the subsequent two to four weeks [32,45]. Advanced cancer patients receiving palliative care often have a poor performance status, are approaching the end of life, use assistive devices, and rely on a caregiver to transport them to their appointments. Patients with elevated symptom distress can receive a high-quality assessment and management of symptoms from the comfort of their home using telemedicine, rather than making an arduous trip to the clinic and facing significant challenges in receiving similar assistance [23]. Caregivers can join telemedicine visits even when not present with the patient, allowing care to be more patient and family-centered. Moreover, telemedicine can help family caregivers maintain their employment and avoid financial implications by decreasing the number of missed workdays needed to accompany their loved ones to in-person appointments. There are also substantial cost savings associated with telemedicine for advanced cancer patients, their families, and health care systems [17]. Telemedicine can also help facilitate collaboration between the specialty oncology and palliative care services and local primary care providers, improving overall patient care [46].

While many aspects of telemedicine may serve as a challenge to the demonstration of empathy by a provider, there are many opportunities that may develop to help foster better knowledge and understanding of the patient. Gurp et al. set out to explore the role that telemedicine can have in facilitating relationships and fostering empathy through exploring palliative care outpatients’ views [47]. In their interviews, patients felt that telemedicine brought about a new level of intimacy into the patient–physician relationship, as the physician could now have a view into the personal environment of the patient [47]. They felt that verbal and non-verbal signals could also add to empathy via telemedicine [47]. It allowed for visits to those who could not leave their beds due to weakness and broke down barriers of distance that existed in the past [47]. Along with appropriate webside manner (Table 1), these other benefits of telemedicine can add to the demonstration of empathy from the physician during their encounters.

## 8. Conclusions

Advanced cancer patients receiving palliative care often have difficulty making it to in-person outpatient visits due to mobility issues, dependency on the caregiver, and high symptom burden. These patients need short interval follow-up appointments to adequately manage the severity of their symptoms, changing or new symptoms, and receive psychological support for themselves and their families. Telemedicine offers a unique and innovative health care delivery platform providing high-quality and timely palliative care to patients with advanced cancer without subjecting them to the challenges associated with in-person visits. Several barriers exist in the implementation of telemedicine worldwide, including lack of training on conducting virtual visits, technology barriers, regulatory barriers, reimbursement issues, managing nonmedical opioid use, etc. Telemedicine can improve access to palliative care for cancer patients. Along with minimizing barriers, more research is needed to improve access to telemedicine and improve the delivery of palliative care via telemedicine in cancer patients.

## Figures and Tables

**Figure 1 cancers-14-01884-f001:**
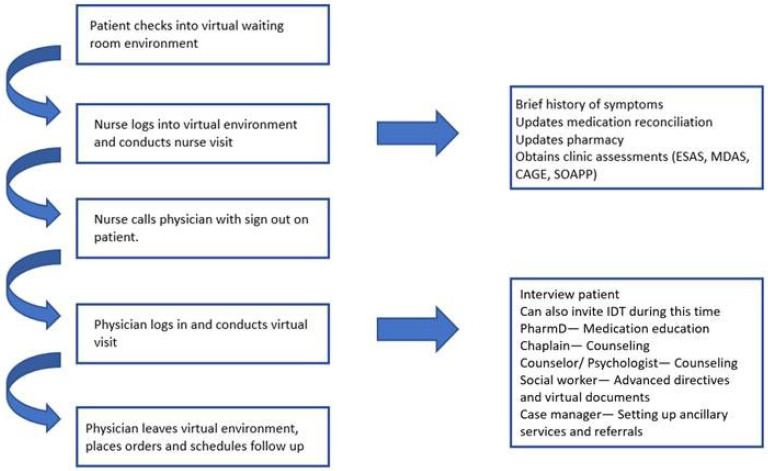
Virtual visit workflow in a supportive care clinic. Abbreviations: ESAS: Edmonton Symptom Assessment System; MDAS: Memorial Delirium Assessment Scale; CAGE: Cut-Down, Annoyed, Guilty and Eye-Opener Questionnaire; SOAPP: The Screener and Opioid Assessment for Patients with Pain; IDT: interdisciplinary team.

**Table 1 cancers-14-01884-t001:** Key Elements and Components of Webside Manner Skills.

Key Element	Components
Proper set up	Quiet environment with minimal potential for disruptions Professional backdrop
	Test platform before first virtual visit
	Body position
	Neutral relaxed posture
	Head and one-third of upper torso should be visualized
	Maintain eye contact
	Camera at eye level
	Situate patient’s onscreen image adjacent to the camera
Acquainting the participant	Wave hello at the start of the visit
	Name the dilemma with the participant
	New or awkward format
	Unexpected disruptions and ambient noise may occur
	Check in: ‘‘How can I make this experience better?’’
Maintaining conversation rhythm	Avoid prolonged silence. Thoughtful brief pauses are favored. Minimize overtalking
	Avoid saying ‘‘mm-hmm.’’ Gently nod instead.
Responding to emotion (e.g., sadness)	Focus on verbal responses ‘‘I wish.’’ ‘‘Take your time. I am here.’’ Consider nonverbal responses
	Lean in slightly to convey intentional listening
	Nod gently
	Place hand over heart to convey empathy
Other considerations	Use phone when there are:
	Persistent technical difficulties
	Participants who either do not have access to the requisite technology or find the virtual visit platform too technically challenging to navigate
	Patients who are too ill to participate
	Non-English-speaking patients who require interpreters: Consider using a virtual visit platform that possesses interpreter services, or use the video platform to visualize the patient and use a separate interpreter phone service for audio
Closing the visit	Summarize the visit
	Verify participant understanding
	Provide opportunity for the participant to voice thoughts, questions, or concerns
	Outline next steps based on goals of care conversation

* This table was obtained from reference [25], Chua et al., with copyright permission from Mary Ann Liebert, Inc. Publishers.

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
