# Peer review of "Telemedicine and Its Past, Present, and Future Roles in Providing Palliative Care to Advanced Cancer Patients"

_cancers, 2022, doi:10.3390/cancers14081884_

Round 1
Reviewer 1 Report
The study highlights the importance of Telemedicine in the management of cancer patients in palliative care, especially after the onset of the Covid-19 pandemic.
I only have one question for the authors:
Among the barriers you have indicated, the impact of Telemedicine on the empathy should be discussed more. Empaty with the patient, face-to-face, represents a fundamental aspect in the doctor-patient relationship, in particular in home palliative care settings for patients with advanced cancer no longer susceptible to oncological therapy, in which the maintenance of an adequate quality of life represents the primary aim of home care. Are there any studies or experiences of the impact of Telemedicine on empaty and quality of life in home palliative care settings?
There are two small errors that need to be corrected:
In the "Webside Manner" section, Table 2 (in the text) is Table 1.
In the legend of Table 1, reference 24 is 25.
Author Response
Thank you for your valuable comment. We have now added a paragraph on page 9 and an additional reference to address this.
“While many aspects of telemedicine may serve as a challenge to the demonstration of empathy by a provider, there are many opportunities that may develop to help foster better knowledge and understanding of the patient. Gurp et al. set out to explore the role that telemedicine can have in facilitating relationships and fostering empathy through exploring palliative care outpatient views.47 In their interviews, patients felt that telemedicine brought about a new level of intimacy into the patient-physician relationship, as the physician could now have a view into the personal environment of the patient.47 They felt that the verbal and non-verbal signals could also add to empathy via telemedicine.47 It allowed for visits to those who could not leave their beds due to weakness and broke down barriers of distance that existed in the past.47 Along with appropriate webside manner (Table 1), these other benefits of telemedicine can add to the demonstration of empathy from the physician during their encounters.”
Reviewer 2 Report
This paper has shows no scientific method, nor a methodology section which describes the nature of the review and the scientific value of the papers reported.
Please add a methods section describing which search string you used, which research questions were answered en how papers were selected following the publication criteria for review studies.
Author Response
Thank you for your thoughtful suggestion. Our article is not a systematic review. It is a narrative review like these 2 other review articles which are part of the same special publication as ours. https://www.ncbi.nlm.nih.gov/pmc/articles/PMC8656500/
https://www.mdpi.com/2072-6694/14/4/1047/htm
We will add information to the abstract and introduction in pages 1 and 2, respectively, to clarify further that this is a narrative review, and match the description used in the articles above to be consistent with others in this special publication, Palliative and Supportive Care in Oncology: An Update. https://www.mdpi.com/journal/cancers/special_issues/Palliative_Supportive_Care_Oncology
“This narrative review article sets out to describe the past and present use of telemedicine in the field of palliative care along with discussing challenges that lay ahead for its continued use. Specifically, we shall discuss telemedicine use before COVID-19 pandemic, the enormous increase in use during the pandemic with emphasis on palliative care, various challenges related to technology, regulations, reimbursement, conveying empathy, and managing non-medical opioid use, and conclude with focusing on the benefits of telemedicine in advanced cancer patients receiving palliative care.”
Reviewer 3 Report
This paper aims to review the use of telemedicine in the field of palliative care in providing care to patients with advanced cancer, clinical benefits , barriers and risks. The topic is of major interest with a huge impact in palliative care daily practice. The english language and style are of very good quality.
Here are a few comments :
The title : I think this title reflects better the content of your paper « Telemedecine use to provide palliative care to ad-vanced cancer patients » The use of télémedecine in palliative care does not seem to be special ; or if you think it does, you should better explain it
Chapter 3 : Telemedicine and its role in palliative medicine
Could you add this paper and discuss it - Hoek et al. BMC Medicine (2017) 15:119 DOI 10.1186/s12916-017-0866-9 – as the result of this unique randomised trial is negative.
Chapter 5 : Barriers in Telemedicine
The Table that you cite in the texte is table 1 ( there is no table 2)
Author Response
Thank you for your suggestion. We have now made the title a little catchier.
“Telemedicine and its past, present, and future roles in providing palliative care to advanced cancer patients”
Chapter 3 : Telemedicine and its role in palliative medicine Added paragraph in this chapter and new reference, ref 20
Could you add this paper and discuss it - Hoek et al. BMC Medicine (2017) 15:119 DOI 10.1186/s12916-017-0866-9 – as the result of this unique randomised trial is negative.
Thank you for your suggestion. We added the reference and a paragraph describing it in page 4.
“Not all studies have shown a positive effect with telemedicine in a palliative care population. Hoek et al. examined patient reported symptoms by comparing a telemedicine intervention of weekly visits versus usual care, defined as routine visits determined by the complexity of the patient’s symptoms and stage of their underlying condition.20 Those who received weekly visits reported a higher symptom burden compared to usual care based on the Edmonton Symptom Assessment Scale.20 The authors felt that weekly visits and frequent assessments might inadvertently have caused increased attention to symptoms and a recall bias among the intervention group.20 These findings must be explored further in future studies.”
Chapter 5 : Barriers in Telemedicine
The Table that you cite in the texte is table 1 ( there is no table 2)
Thank you. We rectified these errors.
Round 2
Reviewer 2 Report
The methods are now described. You could have briefly discussed how you selected the papers in this review. For a narrative review it is recommended, not obligatory.
I would like you to add 'narrative review' to the title.